# Ionic Dioxidovanadium(V) Complexes with Schiff-Base Ligands as Potential Insulin-Mimetic Agents—Substituent Effect on Structure and Stability

**DOI:** 10.3390/molecules27206942

**Published:** 2022-10-16

**Authors:** Anna Jurowska, Janusz Szklarzewicz, Maciej Hodorowicz, Wiktoria Serafin, Ennio Zangrando, Ghodrat Mahmoudi

**Affiliations:** 1Faculty of Chemistry, Jagiellonian University, Gronostajowa 2, 30-387 Kraków, Poland; 2Department of Chemical and Pharmaceutical Sciences, University of Trieste, Via L. Giorgieri 1, 34127 Trieste, Italy; 3Department of Chemistry, Faculty of Science, University of Maragheh, Maragheh P.O. Box 55181-83111, Iran

**Keywords:** vanadium, ionic complexes, Schiff base ligand, crystal structure, UV–Vis spectra

## Abstract

Four dioxidovanadium(V) complexes with Schiff-base ligands based on 2-hydroxybenzhydrazide with four different substituted salicylaldehydes (5-chlorosalicylaldehyde, 3,5-dichlorosalicylaldehyde, 5-nitrosalicylaldehyde, 3-bromo-5-chlorosalicylaldehyde) were synthesized and described, by using V_2_O_5_ and triethylamine. The single crystal X-ray structure measurements as well as elemental analyses and IR spectra confirmed the formulas of the ionic complexes with a protonated triethylamine acting as counterion, HTEA[VO_2_(L)] (HL = Schiff-base ligand). The kinetic stability of the complexes at pH = 2 and 7 was discussed with respect to the neutral vanadium(V) complexes previously studied as potential insulin-mimetic agents. A correlation between the substituents in an aromatic ring of the Schiff-base ligands with crystal packing, and also with the stability of the compounds, was presented.

## 1. Introduction

Vanadium compounds are widely investigated because of their biological role including insulin-mimetic, anticancer, anti-inflammatory, or antibacterial activity [1,2,3,4,5,6,7]. Similarly, Schiff-base ligands have potential application in pharmacotherapy in view of their antidiabetic, antimicrobial, and anticancer activity [8,9,10]. The role of vanadium in insulin-mimetic compounds for type I and type II diabetes treatment was recognized in 1899 and intense studies were started in the 1990s. In this field the bis(maltolato)oxidovanadium(IV) (BMOV), and its bis(ethylmaltolato)oxido-vanadium(IV) (BEOV) analogue have been the most intensively studied compounds, which were used in clinical tests [11,12,13]. Our previous studies focused on the biological activity of the vanadium Schiff-base complexes, where Schiff bases were composed of aldehydes and relatives of amines—hydrazides, to form hydrazone type ligands. These complexes can be potentially used as insulin-mimetic compounds. The vanadium complexes with Schiff bases obtained up until now have manifested problems—most often for their cytotoxicity, very low solubility in water, instability at pH = 2 and in transport to cells, and, last but not least, difficulties in determining the crystal structure. Therefore, it is worth searching for new organic vanadium compounds in order to optimize their pharmaceutical activity. So far, we have tested plenty of vanadium(III, IV and V) complexes with Schiff bases, controlling both the starting compound for the synthesis of complexes, such as [VO(acac)_2_], VOSO_4_, [V(acac)_3_], and V_2_O_5_, as well as changing the substituents in the aromatic ring of aldehyde and hydrazide—Schiff-base components [14,15,16,17,18,19,20]. The obtained compounds were highly soluble in organic solvents, but insoluble in water. Thus, we used a DMSO–H_2_O solvent mixture to test the stability of the complexes. In our last publication, we described the synthesis of a vanadium(V) complex with triethylamine (TEA) as countercation—HTEA[VO_2_(L)] (where L = Schiff base formed by the reaction of 5-bromosalicylaldehyde with 2-hydroxybenzhydrazide) [21]. In the formed ionic complexes, we were able to obtain single crystals and determine the molecular structure where a hydrogen interaction appears between the cation and the complex anion. The ionic structure of such compounds should increase the solubility of the complexes in water. Moreover, the ionic vanadium(V) complexes with Schiff-base ligands show antibacterial and anticancer activities [22,23]. In this work, we present a series of vanadium(V) complexes with hydrazone Schiff-base ligand-type, and protonated triethylamine as cation, along with the structural and physicochemical characterization. In particular, the influence of substituents in the Schiff-base components on the structure and stability of complexes at pH = 2 and 7 was investigated.

## 2. Experimental

### 2.1. Materials and Methods

V_2_O_5_, triethylamine (TEA), 2-hydroxybenzhydrazide, 5-chlorosalicylaldehyde, 3,5-dichlorosalicylaldehyde, 5-nitrosalicylaldehyde, 3-bromo-5-chlorosalicylaldehyde were of analytical grade (Aldrich) and were used as received. Ethanol (98%) of pharmaceutical grade was from Polmos (Poland), all other solvents were of analytical grade and were used as supplied. Microanalysis of carbon, hydrogen and nitrogen was performed using Elementar Vario MICRO Cube elemental analyzer. Electronic absorption spectra were recorded with a Shimadzu UV-3600 UV-vis-NIR spectrophotometer equipped with a CPS-240 temperature controller. IR spectra were recorded on a Nicolet iS5 FT-IR spectrophotometer.

### 2.2. Syntheses of HTEA[VO_2_(L_n_)] (***1***–***4***)

All the compounds were obtained as result of the following synthetic procedure: 1 mmol of 2-hydroxybenzhydrazide and 1 mmol of the respective salicylaldehyde, namely 5-chloro-salicylaldehyde, 3,5-dichloro-salicylaldehyde, 5-nitro-salicylaldehyde, 3-bromo-5-chloro-salicylaldehyde (as indicated in Table 1), were refluxed in 50 mL of methanol for 15 min. To the resulting yellow solution, 0.5 mmol of V_2_O_5_ was added and the mixture was refluxed for additional 5 min. The color of the solution changed to dark red. Then, 2 mL of triethylamine (TEA) was added and the mixture was refluxed for additional 5 min and left for crystallization. After several days, crystals suitable for X-ray diffraction measurement were taken directly from the solution. The residual of the precipitate formed was filtered off, washed with methanol, and dried in air. The yield of individual syntheses was almost quantitative (close to 85–95%). The formulas and elemental analysis results of the diamagnetic complexes are given in Table 1.

### 2.3. Crystallographic Data Collection and Structure Refinement

Diffraction intensity data for compounds **1**–**4** were collected on a Rigaku XtaLAB Synergy-S diffractometer with mirror-monochromated Cu-Kα radiation (λ = 1.54184 Å). Cell refinement and data reduction were performed using CrysAlisPro firmware [24]. The positions of all non-hydrogen atoms were determined by direct methods using SHELXL-2019/2 [25,26]. All non-hydrogen atoms were refined anisotropically using weighted full-matrix least-squares on F^2^, and refinements were carried out using SHELXL-2019/2 [25,26]. All hydrogen atoms were positioned at idealized positions, except those of the OH group and NH of protonated trimethylamine, which were freely refined. 

The crystal data and structure refinement parameters for complexes **1**–**4** are collected in Table 2. 

CCDC 2161295, 2161337, 2161365 and 2161385 contain the supplementary crystallographic data for **1**–**4**, respectively. These data can be obtained free of charge from The Cambridge Crystallographic Data Centre via www.ccdc.cam.ac.uk/data_request/cif accessed on 26 September 2022. 

The XRD powder diffractograms of **2**–**4** (crystallinity tests) were recorded at 294 K on a Philips X’Pert-Pro diffractometer equipped with a Ni-filtered Cu-Kα radiation (λ = 1.54059 Å) over 2θ range from 5 to 50° with a counting time of 1 s and a step size of 0.04°. The X-ray source was operated at 30 mA and 40 kV.

## 3. Results and Discussion

### 3.1. Crystal Structure Description

The X-ray structural analyses show that complexes **1**–**4** have a close comparable structure and, in particular, complexes **1** and **3** present isomorphous structures and similarly **2** and **4**, as can be evidenced from the unit cell parameters (Table 2). An ORTEP view of the crystallographic independent unit for all complexes is shown in Figure 1, and a selection of bond lengths and angles is reported in Table 3. The XRD powder diffractograms of **2**–**4** are presented in Appendix A and show that the polycrystalline sample is similar to the monocrystalline one.

In each anionic [VO_2_(L)]^−^ complex, the vanadium atom is pentacoordinated by two oxido groups and by the anionic tridentate chelating ONO Schiff-base ligand through the phenoxo oxygen O1, the imino nitrogen N1 and the carboxylate oxygen O2. The V-O1 bond distances that fall in the narrow range 1.9874(10)–1.9899(15) Å are systematically shorter by ca. 0.07 Å with respect to V-O2 ones (range 1.9052(10)–1.9264(15) Å). The V-N1 bond lengths vary from 2.1391(17) to 2.1740(13) Å. The V=O4 bond distances (of oxo atom involved in hydrogen bond with HTEA, see below) have a mean value of 1.645 Å, systematically slightly longer than the V = O5 bonds that average to 1.618 Å. However, the different substituents on the phenol ring do not seem to affect the electronic properties of the donor atoms and thus the coordination sphere of vanadium.

The vanadium in the isomorphous structures of **1** and **3** exhibits a slightly distorted square–pyramidal coordination sphere, as ascertained by the τ parameter [27] of 0.151 and 0.122, respectively. On the other hand, the metal in **2** and **4** presents a coordination environment in between a square–pyramidal and trigonal bipyramidal (τ = 0.495 and 0.470, respectively, being τ = 0 for an ideal square pyramid and 1 for an ideal trigonal bipyramid). These findings are confirmed also by the program SHAPE 2.1 [28], which indicates a square pyramid geometry for **1** and **3,** while there is a trigonal bipyramid geometry for **2** and **4** (Appendix A).

These bonding parameters are in agreement with those detected in other dioxovanadium complexes [23,29,30]. In all [VO_2_(L)]^−^ anions, a strong intramolecular H-bond is realized between the O3-H and nitrogen N2 with O…N distance of ca. 2.58 Å (Table 4).

In all compounds, the protonated triethylamine is hydrogen bound to oxygen O4 of the vanadium complex, with N3…O4 distance in between 2.686(2)–2.787(2) Å (Figure 1, Table 4).

The crystal packing of compound **1** is shown in Figure 2, where complexes are interacting by π-stacking interactions between phenol rings (centroid-to-centroid distance of 4.185(1) Å. A similar crystal packing is observed for **3**, with slightly longer π-stacking interactions 4.411(1) Å. The crystal packing of compounds **2** and **4** exhibits pairs of complexes centrosymmetrically related to form reciprocal π-stacking interactions with centroid distances of 3.606(1) and 3.656(1) Å, respectively (Figure 3). Finally, numerous unconventional hydrogen bonds of type C-H…O, C-H…Cl and C-H…Br, as indicated in Table 4 and Appendix A, as well as C-H…π interactions (Table 5) reinforce the crystal packing of all the complexes.

### 3.2. Spectroscopic Characteristics

The IR spectra of **1**–**4** were measured in the 400–4000 cm^−1^ range and presented in Figure 4. As the complexes differ only in the substituent in the fragment derived from the aldehyde, the spectra are very similar. The position of ν_(V=O)_ stretching frequency is 939, 942, 936, 944 cm^−1^ (for **1**–**4**, respectively). For **1** and **3** as well as for **2** and **4** the position is very similar, as these pairs of compounds show an isomorphous structure. These bands were also observed in other dioxido vanadium compounds [31,32,33]. The bands related to the stretching frequencies of -C=N are observed in the range of 1610–1630 cm^−1^. The stretching vibration of the -OH group in the Schiff-base ligand is observed at ca. 2980 cm^−1^, and that of the -NH vibration in the cation at 2450–2600 cm^−1^.

The UV-Vis spectra of all the complexes were recorded in EtOH, MeOH, MeCN, DMSO and water (see Figure 5).

The solubility of **1**–**4** in different organic solvents was very good, while in water the complexes exhibited partial solubility that increased with increasing temperature. The complexes **1**, **2** and **4** show intense bands in the range of 400–410 nm, which can be assigned to ligand-to-metal charge transfer (LMCT). Compound **3** shows this band at higher energy (ca. 380–390 nm) and presents also a well-formed band at ca. 308 nm, being the only compound that does not have a halogen atom as a substituent. The stability of the complexes was proved in DMSO (Figure 6), in water (pH ca. 7) as well as in 0.01 M HCl (pH ca. 2, imitating the environment in the stomach). All measurements were performed at 37 °C. As the compounds are stable in DMSO, this is an important feature, since this solvent can be used as a solvent to transport molecules across cell membranes like the skin. Since these compounds were soluble in water, we had the opportunity to determine for the first time their stability in water. By using a DMSO/H_2_O mixture, the complexes were shown to be stable at pH = 7, but were unstable at pH = 2 [16,17,18,19,20]. In Figure 7 and Figure 8 a similar behavior can be observed: at pH = 7 the band at 400 nm does not change with time (Figure 7). Acidification of the solution causes the formation of a suspension (raised background in the entire range as shown in Figure 8). The disappearance of the bands at ca. 400 nm, combined with an increase in the intensity of the bands at ca. 300 nm, suggests the decomposition of Schiff bases into components. 

## 4. Conclusions

Four complexes of vanadium(V) with Schiff-base ligands were isolated and structurally characterized. The addition of triethylamine caused deprotonation of the Schiff-base ligand and isolation of ionic complexes with Et_3_NH^+^ as cation. The structural data indicate that complexes **1** and **3** have an isomorphous structure, and similarly **2** and **4**. These complexes differ only in the number and type of substituents on the ring constituting the part of the aldehyde component of the Schiff-base ligand. The introduction of an additional substituent on the salicylaldehyde ring induces a decrease in the symmetry of the unit cell for complexes **2** and **4**, which crystallize in the triclinic system, in contrast to compounds **1** and **3** crystallizing in the monoclinic system. The present ionic complexes display a better solubility in water with respect to similar neutral complexes of vanadium(V) previously reported, and this represents an important aspect from the point of view of their potential biological applications. Complex **2** showed the best solubility, due to the additional chlorine substituent. The stability of complexes in DMSO and water is very good; however, they are unstable at pH = 2. Research on improving compound stability in the acidic environment by changing the cation is in progress.

## Figures and Tables

**Figure 1 molecules-27-06942-f001:**
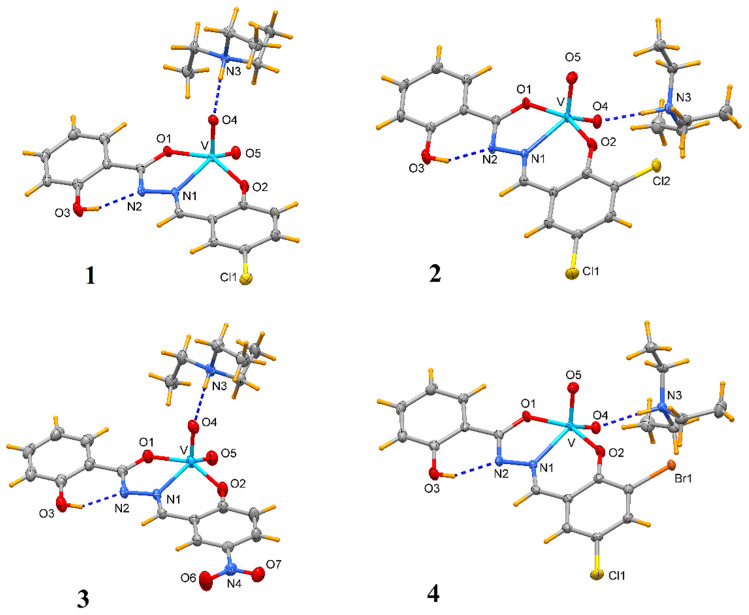
Molecular structure of complexes **1**–**4** (ellipsoid probability at 50%).

**Figure 2 molecules-27-06942-f002:**
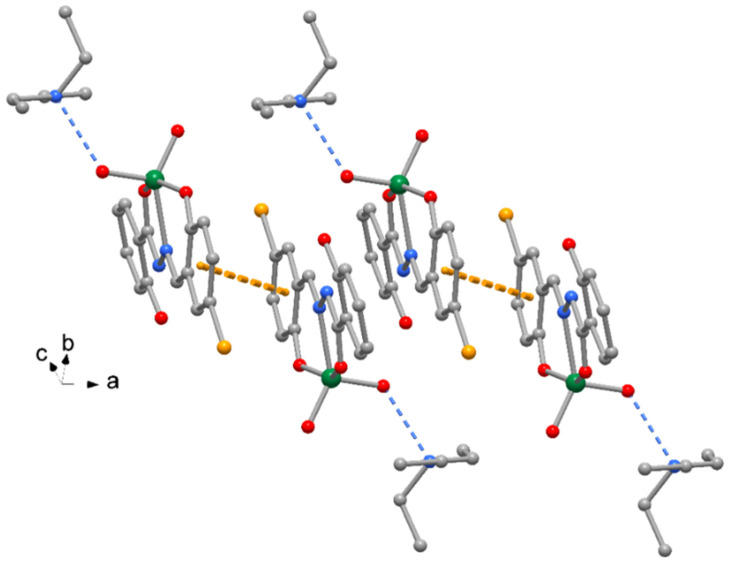
Crystal packing of complex **1** with indication of π-stacking between complex pairs with indication of the orientation with respect to cell axes (orange dotted lines = π-stacking interactions, blue dotted lines = H-bonds). The same crystal packing is exhibited by complex **3**.

**Figure 3 molecules-27-06942-f003:**
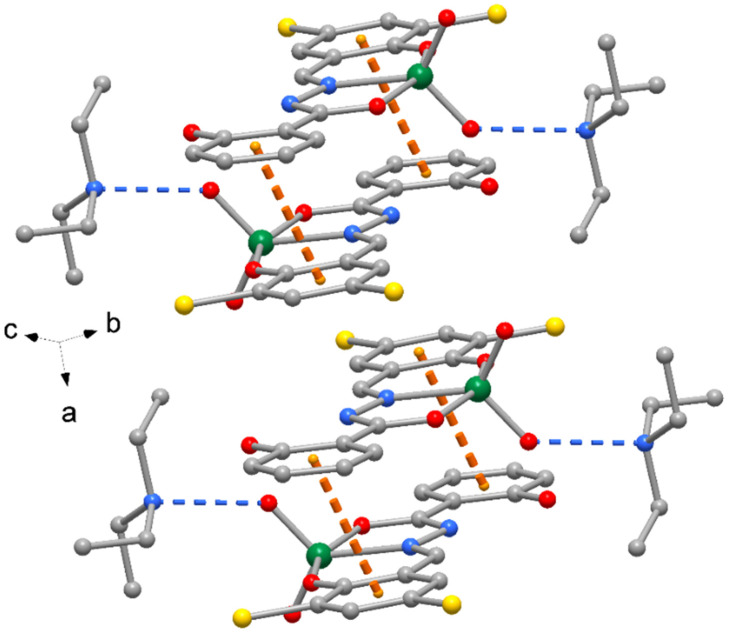
Crystal packing of complex **2** with indication of π-stacked centrosymmetric complex pairs with indication of the orientation with respect to cell axes (orange dotted lines = π-stacking interactions, blue dotted lines = H-bonds). The same crystal packing is exhibited by complex **4**.

**Figure 4 molecules-27-06942-f004:**
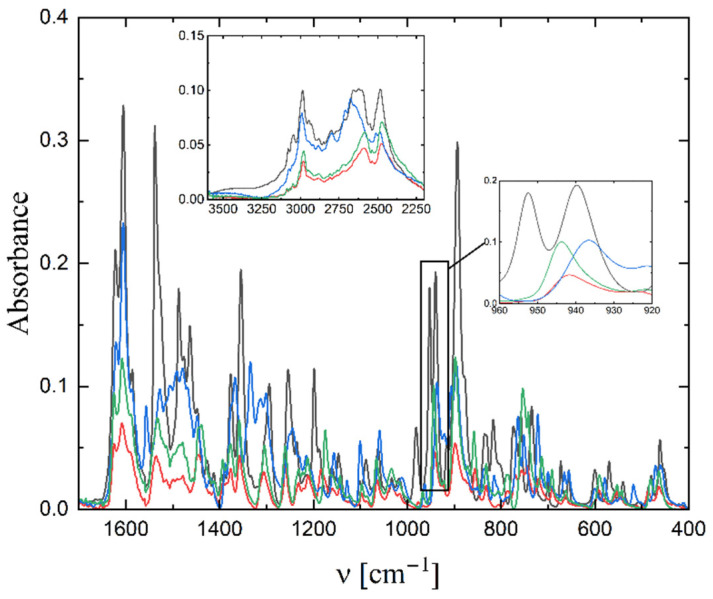
IR spectra of complex **1** (black line), **2** (red), **3** (blue) and **4** (green).

**Figure 5 molecules-27-06942-f005:**
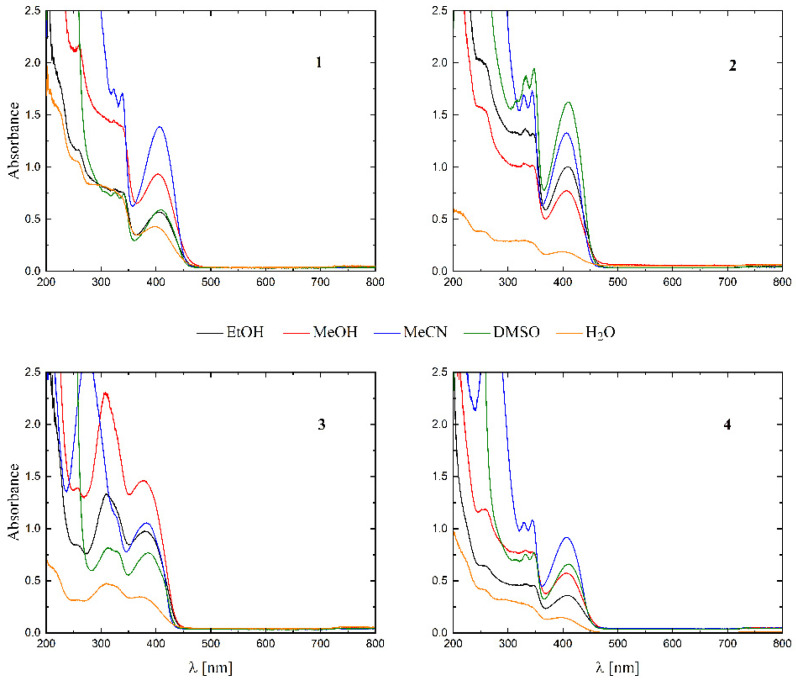
UV-Vis spectra of complexes **1**–**4** in various solvents (black—EtOH, red—MeOH, blue—MeCN, green—DMSO and orange—H_2_O).

**Figure 6 molecules-27-06942-f006:**
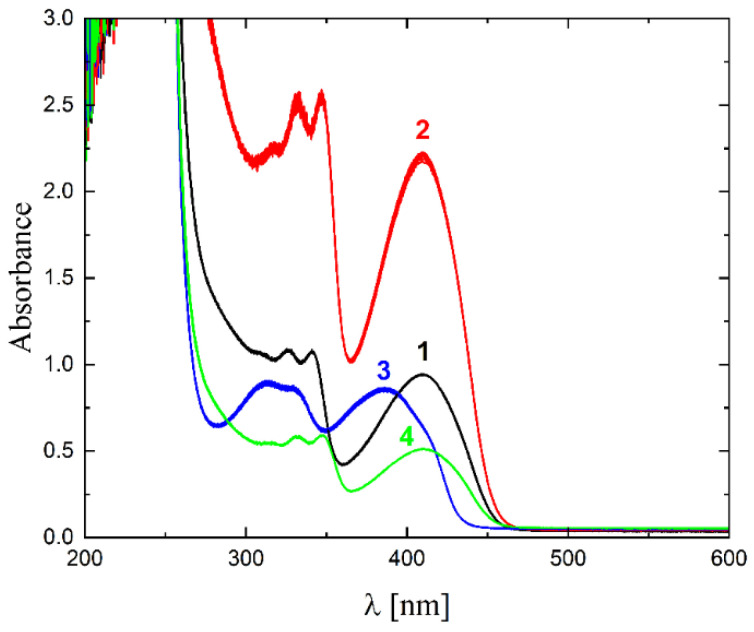
UV-Vis spectra of complexes **1**–**4** in DMSO (black, red, blue and green lines, respectively), T = 37 °C, d = 1 cm, 15 spectra measured in 780 s intervals.

**Figure 7 molecules-27-06942-f007:**
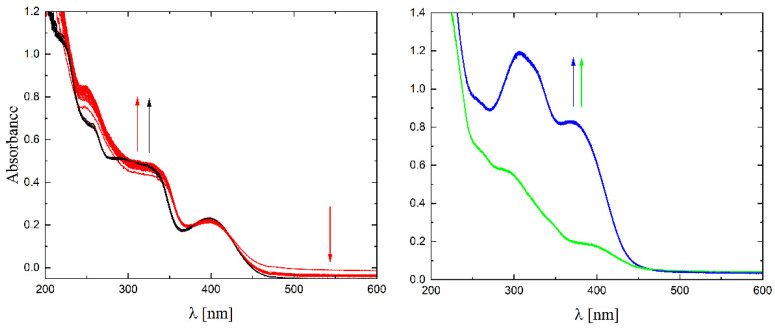
UV-Vis spectra of complexes **1**–**4** in water (pH ≈ 7) (black, red, blue and green lines, respectively), T = 37 °C, d = 1 cm, 15 spectra measured in 360 s intervals. The arrows show direction of changes.

**Figure 8 molecules-27-06942-f008:**
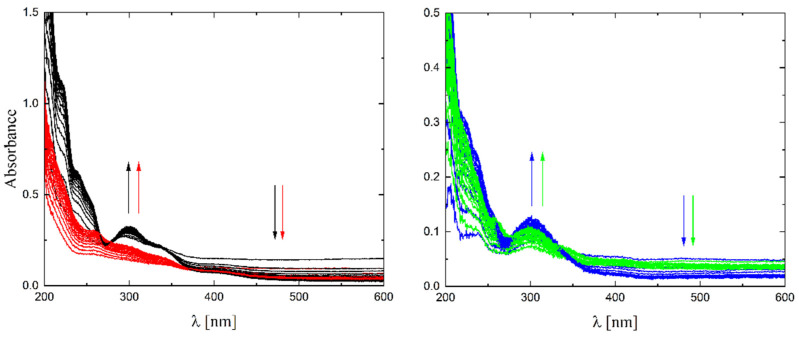
UV-Vis spectra of complexes **1**–**4** (black, red, blue and green lines, respectively) in 0.01M HCl (pH ≈ 2) T = 37 °C, d = 1 cm, 15 spectra measured in 360 s intervals. The arrows show direction of changes.

**Table 1 molecules-27-06942-t001:** The formulas of compounds **1**–**4** with elemental analysis results and components of Schiff-base ligands used in synthesis.

	Complex Formula	Elemental Analysis [%] exp calc	Hydrazide	Aldehyde	Ligand L_n_ Formula
**1**	HTEA[VO_2_(L_1_)]	C, 50.96; 50.59 H, 5.21; 5.52 N, 8.71; 8.85	2-hydroxy-benzhydrazide	5-chloro-salicylaldehyde	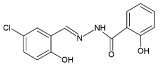
**2**	HTEA[VO_2_(L_2_)]	C, 47.31; 47.17 H, 4.90; 4.95 N, 8.24; 8.25		3,5-dichloro-salicylaldehyde	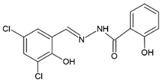
**3**	HTEA[VO_2_(L_3_)]	C, 49.55; 49.49 H, 5.51; 5.40 N, 11.49; 11.54		5-nitro-salicylaldehyde	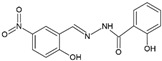
**4**	HTEA[VO_2_(L_4_)]	C, 43.30; 43.38 H, 4.46; 4.55 N, 7.43; 7.59		3-bromo-5-chloro-salicylaldehyde	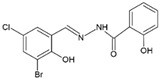

**Table 2 molecules-27-06942-t002:** Crystal data and details of structure refinement for complexes **1**–**4**.

	1	2	3	4
Empirical formula	C_20_H_25_ClN_3_O_5_V	C_20_H_24_Cl_2_N_3_O_5_V	C_20_H_25_N_4_O_7_V	C_20_H_24_BrClN_3_O_5_V
Formula weight	473.82	508.26	484.38	552.72
Crystal system	Monoclinic	Triclinic	Monoclinic	Triclinic
Space group	*P* 2_1_/n	*P* $\bar{1}$	*P* 2_1_/n	*P* $\bar{1}$
a [Å]	7.46809(3)	7.2268(2)	7.4272(1)	7.3085(1)
b [Å]	13.00924(4)	11.4466(2)	13.1272(1)	11.4480(1)
c [Å]	21.59664(7)	13.9949(2)	21.9821(1)	14.2515(2)
α [°]	90	106.228(1)	90	108.525(1)
β [°]	99.2988(3)	101.140(2)	97.826(1)	102.008(1)
γ [°]	90	91.472(2)	90	90.791(1)
Volume [Ǻ^3^]	2070.632(13)	1086.71(4)	2123.26(3)	1101.75(2)
Z	4	2	4	2
Density [mg/m^3^]	1.520	1.553	1.515	1.666
μ [mm^−1^]	5.524	6.411	4.358	7.332
F(000)	984	524	1008	560
θ range [^o^]	3.98–75.68	3.36–75.30	3.93–75.08	3.36–75.41
Index ranges	−9 ≤ h ≤ 8	−7 ≤ h ≤ 8	−9 ≤ h ≤ 7	−8 ≤ h ≤ 8
	−15 ≤ k ≤ 16	−14 ≤ k ≤ 14	−16 ≤ k ≤ 16	−14 ≤ k ≤ 14
	−26 ≤ l ≤ 27	−17 ≤ l ≤ 17	−26 ≤ l ≤ 26	−17 ≤ l ≤ 17
Reflections collected	73160	37876	73570	38590
Indep. reflections	4167	4273	4202	4326
R_int_	0.0456	0.0704	0.0717	0.0552
Parameters	282	287	300	291
GOF on *F*^2^	1.041	1.175	1.124	1.092
*R*1 [*I* > 2σ(*I*)]	0.0273	0.0403	0.0376	0.0310
*wR*2 [*I* > 2σ(*I*)]	0.0761	0.1135	0.1044	0.0854
*R*1 (all data)	0.0273	0.0422	0.0383	0.0314
*wR*2 (all data)	0.0761	0.1174	0.1050	0.0858
residuals [e·Ǻ^−3^]	0.355, −0.513	0.535, −0.912	0.334, −0.639	0.565, −0.900

**Table 3 molecules-27-06942-t003:** Selected bond lengths (Å) and angles (°) for complexes **1**–**4**.

	1	2	3	4
V-O(1)	1.9874(10)	1.9889(14)	1.9752(12)	1.9899(15)
V-O(2)	1.9052(10)	1.9205(15)	1.9151(13)	1.9264(15)
V-N(1)	2.1623(11)	2.1432(17)	2.1740(13)	2.1391(17)
V-O(4)	1.6477(10)	1.6448(16)	1.6419(14)	1.6463(15)
V-O(5)	1.6256(10)	1.6145(15)	1.6192(13)	1.6147(15)
O(1)-V-O(2)	150.02(4)	155.98(6)	149.31(6)	156.09(6)
O(1)-V-N(1)	73.35(4)	74.04(6)	73.28(5)	74.04(6)
O(1)-V-O(4)	100.16(5)	95.99(7)	100.42(6)	94.91(7)
O(1)-V-O(5)	93.15(4)	95.65(7)	94.18(6)	95.83(7)
O(2)-V-N(1)	81.73(4)	81.94(6)	81.38(5)	82.11(6)
O(2)-V-O(4)	103.88(5)	98.55(7)	104.09(6)	98.44(7)
O(2)-V-O(5)	95.96(5)	97.44(7)	95.31(6)	98.34(7)
O(4)-V-N(1)	109.94(5)	126.25(7)	109.13(6)	127.86(7)
O(5)-V-N(1)	140.96(5)	124.02(7)	141.97(6)	122.33(7)
O(4)-V-O(5)	108.42(5)	109.30(8)	108.41(7)	109.25(8)
C-O(1)-V	119.08(8)	118.19(13)	119.53(10)	118.21(13)
C-O(2)-V	135.12(9)	138.64(14)	135.45(12)	138.70(14)
C-N(1)-V	129.70(9)	130.69(14)	129.42(11)	130.71(14)
N(2)-N(1)-V	115.55(8)	115.14(12)	115.23(10)	115.40(12)

**Table 4 molecules-27-06942-t004:** Selected hydrogen bonds for complexes **1**–**4** [Å/°].

D-H…A	d(D-H)	d(H…A)	d(D…A)	<(DHA)
**1**				
N(3)-H(3n)…O(4)	0.88(2)	1.85(2)	2.7293(15)	176.2(19)
O(3)-H(3o)…N(2)	0.83(3)	1.83(3)	2.5857(15)	150(2)
**2**				
N(3)-H(3n)…O(4)	1.00	1.71	2.706(2)	178.6
O(3)-H(3o)…N(2)	0.87(4)	1.78(4)	2.572(2)	150(3)
**3**				
N(3)-H(3n)…O(4)	0.81(2)	1.98(3)	2.787(2)	172(2)
O(3)-H(3o)…N(2)	0.83(3)	1.88(3)	2.6059(18)	146(3)
**4**				
N(3)-H(3n)…O(4)	0.91(3)	1.77(3)	2.686(2)	176(3)
O(3)-H(3o)…N(2)	0.78(4)	1.86(4)	2.567(2)	150(4)

**Table 5 molecules-27-06942-t005:** The X-H∙∙∙Cg(π) interaction in **1** and **3** [Å/°].

X-H…Cg(π)	d(H…Cg)	d(X-H…Cg)	d(X…Cg)	Cg1
**1**				
C19-H19b…Cg1	2.76	142	3.5853(16)	C2-C3-C7-C13-C17-C5 at -x, 1-y, -z
**3**				
C26-H26b…Cg1	2.89	136	3.665(2)	C17-C18-C19-C20-C21-C22 at 1-x, 2-y, 1-z

## Data Availability

Samples of the compounds are available from the authors.

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
