# Peer review of "Ionic Dioxidovanadium(V) Complexes with Schiff-Base Ligands as Potential Insulin-Mimetic Agents—Substituent Effect on Structure and Stability"

_molecules, 2022, doi:10.3390/molecules27206942_

Round 1

Reviewer 1 Report

The authors Jurowska and Mhmoudi and al. have submitted a manuscript reporting four new vanadium(V) complexes based on substituted Schiff base ligands, having the general formula HTEA[VO2L]. The four ligands differ from each other by the nature of the substituents (number of substituents, positions on the aromatic ring and electronic properties) on the phenol ring of the ligands. The four complexes have been characterized by spectroscopic techniques (IR and UV-vis absorption), by elemental analyses and by X-Ray diffraction on single crystals. The authors have also studied the stability of the four compounds in water, in acidified water and in DMSO.

The scientific content is of interest for the people working in the field of vanadium(V) coordination chemistry and that of Schiff bases as ligands. The article is well structured with appropriate literature references but we would have appreciated more citations regarding the chemistry of Schiff base vanadium(V) complexes and not only those from the authors, a review maybe? The results are

The experimental section is concise and lack a little bit of precision. The yield for the synthesis (page 2) of the four complexes are missing, the authors have to add this information. The phase purity can also be checked by XRay powder diffraction as the theoretical powder diffractogram can be extracted from the crystal structure, a comparison between the experimental and theoretical diffractograms would be appropriated as the complexes are obtained as crystalline materials and that would complete the microanalysis results. Please, also remove page 2, line 67 and 68, the mention of the magnetic susceptibility measurement method as the complexes are diamagnetic!

In the crystal structure description part, the coordination environment of the vanadium atoms in the four structures are given towards the t parameter, explained from a reference from 1984. The authors could also check the vanadium atoms coordination sphere with the free Shape program, a software for obtaining continuous shape measures of coordination polyhedra and derived properties.

The spectroscopic characterizations section is clear and well described but the quality of the figures is poor, as that for the crystal structure description section, more precisely, the whole resolution of the images is bad, especially the "text" parts.

More generally, the manuscript is rather well written with however some syntax and spelling errors. The conclusions are in adequacy with the experimental part and the description of the results and answer the problematic posed in the introduction of the article.

Author Response

The authors Jurowska and Mahmoudi and al. have submitted a manuscript reporting four new vanadium(V) complexes based on substituted Schiff base ligands, having the general formula HTEA[VO2L]. The four ligands differ from each other by the nature of the substituents (number of substituents, positions on the aromatic ring and electronic properties) on the phenol ring of the ligands. The four complexes have been characterized by spectroscopic techniques (IR and UV-vis absorption), by elemental analyses and by X-Ray diffraction on single crystals. The authors have also studied the stability of the four compounds in water, in acidified water and in DMSO. The scientific content is of interest for the people working in the field of vanadium(V) coordination chemistry and that of Schiff bases as ligands. The article is well structured with appropriate literature references but we would have appreciated more citations regarding the chemistry of Schiff base vanadium(V) complexes and not only those from the authors, a review maybe?

Response: We added the additional references. 

The experimental section is concise and lack a little bit of precision. The yield for the synthesis (page 2) of the four complexes are missing, the authors have to add this information. The phase purity can also be checked by XRay powder diffraction as the theoretical powder diffractogram can be extracted from the crystal structure, a comparison between the experimental and theoretical diffractograms would be appropriated as the complexes are obtained as crystalline materials and that would complete the microanalysis results. Please, also remove page 2, line 67 and 68, the mention of the magnetic susceptibility measurement method as the complexes are diamagnetic!

Response: We added the yield for the syntheses. We measured XRD for 2-4 compounds and we added diffractogram into Supplementary Materials, we also replenished the discussion part. Unfortunately, we could not measure XRD for complex 1 as we used all the sample for other measurements. We removed line 67 and 68 according to the comment. 

In the crystal structure description part, the coordination environment of the vanadium atoms in the four structures are given towards the t parameter, explained from a reference from 1984. The authors could also check the vanadium atoms coordination sphere with the free Shape program, a software for obtaining continuous shape measures of coordination polyhedra and derived properties.

Response: Results of Shape program are included as Supplementary in Table 2S. Following data of tau values, a sentence was included in the text: “These findings are confirmed also by program SHAPE 2.1 [28], which indicates a square pyramid geometry for 1 and 3 while a trigonal bipyramid for 2 and 4 (Table S2). 

The spectroscopic characterizations section is clear and well described but the quality of the figures is poor, as that for the crystal structure description section, more precisely, the whole resolution of the images is bad, especially the "text" parts.

Response: We improved the figures. 

More generally, the manuscript is rather well written with however some syntax and spelling errors. The conclusions are in adequacy with the experimental part and the description of the results and answer the problematic posed in the introduction of the article.

Reviewer 2 Report

The paper "Ionic dioxidovanadium(V) complexes with Schiff base ligands as potential insulin-mimetic agent – substituent effect on the structure and stability" is devoted to the synthesis and study of several complexes of vanadyl with hydrazones capable of posessing anti-diabetic effect. Authors determined the structure of metal complexes using X-ray analysis, recorded IR-spectra to confirm the donor centers of ligands. They also found metal complex to decompose with time in acidic aqueous solutions. The paper is quite decent, and I have judst a couple of minor comments:

1. The studied ligands are rather arguably hydrazones than Schiff bases although hydrazones are very alike of Schiff bases.

2. Seeing word "stability" in the title, I was very glad because I thought that somebody determines the stability constants of metal complexes at long last. I'd recommed to specify, what exactly do you mean under stability (like 'kinetic stability') in the title and abstract. And, please, consider determining the stability constants of these metal complexes in future - it can be done using even the obtained kinetic data. I believe, the complexation process is reversible, so K = k1/k-1. Other possible ways to determine the stability constant at fixed pH value include spectrophotometric titration of ligand against metal (or vice versa).
Or, maybe, the kinetic process refers to the decomposition of ligand?

3. The increased absorbance at the long-wavelength region of spectra (400 to 600 nm, Fig. 8) concerns me. In my practice, I've seen such changes in UV-Vis spectra whent some precipitation processes occurred during the experiment. It is difficult to see, do absorbance increase or decrease with time; is there some processes of colloid formation or, on the contrary, dissolution?

Author Response

The paper "Ionic dioxidovanadium(V) complexes with Schiff base ligands as potential insulin-mimetic agent – substituent effect on the structure and stability" is devoted to the synthesis and study of several complexes of vanadyl with hydrazones capable of posessing anti-diabetic effect. Authors determined the structure of metal complexes using X-ray analysis, recorded IR-spectra to confirm the donor centers of ligands. They also found metal complex to decompose with time in acidic aqueous solutions. The paper is quite decent, and I have just a couple of minor comments: 1.     The studied ligands are rather arguably hydrazones than Schiff bases although hydrazones are very alike of Schiff bases.

Response: We indicated in the Introduction that Schiff bases are composed of aldehydes and hydrazides and formally give hydrazone type ligands. The changes in text are indicated in green. 

2. Seeing word "stability" in the title, I was very glad because I thought that somebody determines the stability constants of metal complexes at long last. I'd recommend to specify, what exactly do you mean under stability (like 'kinetic stability') in the title and abstract. And, please, consider determining the stability constants of these metal complexes in future - it can be done using even the obtained kinetic data. I believe, the complexation process is reversible, so K = k1/k-1. Other possible ways to determine the stability constant at fixed pH value include spectrophotometric titration of ligand against metal (or vice versa).Or, maybe, the kinetic process refers to the decomposition of ligand?

Response: We are sorry for your disappointment. The complexes in solution, especially in acidic conditions, undergo decomposition and ligand release. In the spectra we do not see isosbestic points, which could indicate a process leading to reach the equilibrium. In manuscript we reefer of course to kinetic stability. At pH = 7, the complexes are stable with time, while at pH = 2, the complexes decompose so fast, that spectral changes are rather related to the Schiff base decomposition into components and not to ligand release. As complex decomposition was of our interest and we do not include the kinetic for ligand decomposition.   

3. The increased absorbance at the long-wavelength region of spectra (400 to 600 nm, Fig. 8) concerns me. In my practice, I've seen such changes in UV-Vis spectra when some precipitation processes occurred during the experiment. It is difficult to see, do absorbance increase or decrease with time; is there some processes of colloid formation or, on the contrary, dissolution?

Response: Absorbance decreases with time at the 400 - 600 nm region. We added for Fig. 8 the additional arrows.